# Methylated Circulating Tumor DNA in Blood as a Tool for Diagnosing Lung Cancer: A Systematic Review and Meta-Analysis

**DOI:** 10.3390/cancers15153959

**Published:** 2023-08-03

**Authors:** Morten Borg, Sara Witting Christensen Wen, Rikke Fredslund Andersen, Signe Timm, Torben Frøstrup Hansen, Ole Hilberg

**Affiliations:** 1Department of Medicine, Vejle Hospital, University Hospital of Southern Denmark, 7100 Vejle, Denmark; morten.hornemann.borg@rsyd.dk (M.B.);; 2Department of Oncology, Vejle Hospital, University Hospital of Southern Denmark, 7100 Vejle, Denmark; 3Department of Regional Health Research, University of Southern Denmark, 5000 Odense, Denmark; 4Department of Biochemistry and Immunology, Vejle Hospital, University Hospital of Southern Denmark, 7100 Vejle, Denmark

**Keywords:** circulating tumor DNA, ctDNA, methylated tumor DNA, liquid biopsy, lung cancer

## Abstract

**Simple Summary:**

Lung cancer is the leading cause of cancer-related deaths and has a poor prognosis. Early detection could improve survival for this large patient group. Certain genes are more frequently changed by methylation in cancer cells compared to healthy cells. This methylated tumor DNA is present in the blood in small quantities and has been suggested as a diagnostic biomarker in many diseases, including lung cancer. The aim of the present literature review was to identify and collate the current evidence on methylated circulating tumor DNA in blood samples as a diagnostic tool for lung cancer. A systematic collection and presentation of the existing evidence will aid future research in this field.

**Abstract:**

Lung cancer is the leading cause of cancer-related deaths, and early detection is crucial for improving patient outcomes. Current screening methods using computed tomography have limitations, prompting interest in non-invasive diagnostic tools such as methylated circulating tumor DNA (ctDNA). The PRISMA guidelines for systematic reviews were followed. The electronic databases MEDLINE, Embase, Web of Science, and Cochrane Library were systematically searched for articles. The search string contained three main topics: Lung cancer, blood, and methylated ctDNA. The extraction of data and quality assessment were carried out independently by the reviewers. In total, 33 studies were eligible for inclusion in this systematic review and meta-analysis. The most frequently studied genes were SHOX2, RASSF1A, and APC. The sensitivity and specificity of methylated ctDNA varied across studies, with a summary sensitivity estimate of 46.9% and a summary specificity estimate of 92.9%. The area under the hierarchical summary receiver operating characteristics curve was 0.81. The included studies were generally of acceptable quality, although they lacked information in certain areas. The risk of publication bias was not significant. Based on the findings, methylated ctDNA in blood shows potential as a rule-in tool for lung cancer diagnosis but requires further research, possibly in combination with other biomarkers.

## 1. Introduction

Lung cancer is the leading cause of cancer-related deaths worldwide [1]. The survival rate of lung cancer patients is strongly correlated with the stage of the disease at diagnosis [2]. Early detection is critical for improving patient outcomes, as it increases the likelihood of successful treatment and long-term survival [3]. Unfortunately, a large proportion of lung cancer cases are diagnosed at an advanced stage [4], and hence, there is a desire for effective screening methods for lung cancer [5]. 

Current screening methods using computed tomography (CT) scans have resulted in a stage shift towards early-stage disease and reduced mortality [6,7]. Low-dose CT screening is now implemented in several countries, among them the US [8,9]. However, several limitations are present, including high costs, radiation exposure, and low adherence to follow-up scans [10,11,12]. Furthermore, the management of screen-detected pulmonary nodules is time-consuming [13] and associated with anxiety and depression among patients [14].

In recent years, there has been a growing interest in the use of circulating tumor DNA (ctDNA) as a non-invasive diagnostic tool for cancer [15,16]. ctDNA is released into the bloodstream by tumor cells and can be detected and quantified through molecular techniques [17]. Many assays are based on polymerase chain reaction (PCR), such as real-time or quantitative PCR and, in recent years, also digital PCR [18]. Other methods for quantitating ctDNA include sequencing techniques such as pyrosequencing or next-generation sequencing (NGS) methods [19]. Methylated ctDNA, which refers to circulating DNA with methyl-groups added to CpGs in a cancer-specific manner, has been shown to be a promising biomarker for cancer diagnosis and prognosis [20].

Several studies have investigated the diagnostic accuracy of methylated ctDNA in the blood of a variety of genes for lung cancer diagnosis [21,22,23]. The results have shown large variations in sensitivity and specificity and also in the choice of biological specimen and assay type [24]. An up-to-date systematic review and meta-analysis of the available evidence is necessary to provide a more comprehensive understanding of the diagnostic performance of methylated ctDNA for the diagnosis of lung cancer.

Hence, the aim of this systematic review and meta-analysis on methylated ctDNA for lung cancer diagnosis is to systematically identify and analyze all relevant studies that have investigated the diagnostic accuracy of methylated ctDNA in blood for lung cancer detection.

In total, 33 studies were eligible for inclusion in this review. The sensitivity and specificity of methylated ctDNA varied across studies, with a summary sensitivity estimate of 46.9% and a summary specificity estimate of 92.9%. The area under the hierarchical summary receiver operating characteristics curve was 0.81.

## 2. Materials and Methods

The review was conducted in accordance with the Preferred Reporting Items for Systematic Reviews and Meta-Analyses (PRISMA) 2020 statement [25]. The study was registered on the Prospero website (registration number CRD42022361536, University of York, York, UK) on 1 October 2022.

### 2.1. Search Strategy and Screening 

The electronic databases MEDLINE, Embase, Web of Science, and Cochrane Library were systematically searched for articles. The search string contained three main topics: (I) Lung cancer, (II) blood, and (III) methylated circulating tumor DNA. All relevant Subject Headings and free text terms were included in each topic and combined with the Boolean operator ‘OR’. The three main topics were combined with the Boolean operator ‘AND’. The detailed search string for each database can be accessed in the Appendix A. We did not include the topic ‘Diagnosis’ in the search because preliminary searches showed that relevant studies would then be omitted. The searches contained no restrictions regarding language, article type, or date of publication since such restrictions could potentially limit study retrieval. The searches were conducted between 16 November 2022 and 12 December 2022. Forward and backward citation searches were performed on all included studies using the Web of Science to extract reference lists and citations. These searches were performed on 24 April 2023.

All references identified from each database were imported into Covidence (Covidence, Melbourne, Australia), with duplicates automatically removed. The initial screening of the title and abstract required only one vote and was performed by one of the three main authors (MB, SW, or OH). 

### 2.2. Eligibility Criteria

Full-text review of all potentially relevant studies was performed independently by two reviewers (MB and SW) in a blinded manner using the Covidence software. In case of disagreement, the study was discussed between the main reviewers until consensus. The inclusion criteria were as follows: (I) Adults diagnosed with lung cancer or undergoing diagnostic work-up for lung cancer; (II) plasma or serum sample collected for quantitative analysis of methylated circulating tumor DNA; (III) tumor cytology or histopathology as a reference standard; (IV) outcome reported as diagnostic sensitivity and specificity or a contingency table with enough data to calculate these diagnostic measures for each reported target. The exclusion criteria were: (I) Case-reports, conference papers, editorials, notes, and literature reviews; (II) studies not in English; (III) pure in silico analyses. 

### 2.3. Data Extraction and Quality Assessment

A data extraction form was developed in Covidence and pilot tested with seven studies. The final data collection form included the first author’s last name, publication year, geographic region, study aims, study design, type of blood sample, analysis method, type of reference standard, number of cases, and number of controls. Study outcomes included gene names, sensitivity and specificity, the area under the curve (AUC), and a contingency table of true positives, false positives, true negatives, and false negatives. Quality assessment was performed using the Quality Assessment of Diagnostic Accuracy Studies 2 (QUADAS-2) tool [26]. 

Both data extraction and quality assessment were performed independently by two reviewers (MB and SW), and any disagreements were discussed until consensus with the option to consult a third reviewer (OH) in case of persisting disagreement. Data extraction issues regarding the analytical methodology were supervised and settled by an expert (RFA).

### 2.4. Statistical Analysis

Data were synthesized in the form of a summary table, including all relevant studies. If diagnostic sensitivity and/or specificity were not reported, these values were calculated using the following formulas: Sensitivity = true positive/(true positive + false negative). Specificity = true negative/(true negative + false positive). The STATA command metandi [27] was used for hierarchical summary receiver operating characteristics (HSROC) plots and for calculating summary effect measures. This method is based on a two-level mixed-effect logistic regression model with independent binomial distribution. The STATA command midas [28] was used for forest plots, including the 95% confidence interval (CI) for each study estimate and for Deek’s funnel plot and corresponding test for assessing the risk of publication bias. The level of significance was set at 0.05. STATA BE version 17 (StataCorp LLC, College Station, TX, USA) was used for statistical analyses. Microsoft Excel (Microsoft Corporation, Redmond, WA, USA) was used for forest plot graphics.

## 3. Results

### 3.1. Search Results and Eligibility

The study selection process is illustrated by a PRISMA flow chart (Figure 1). Searches performed in MEDLINE, Embase, Web of Science, and Cochrane Library resulted in 15,211 records. Forward and backward citation searching identified a further 2425 references. Screening for duplicates removed 5738 records, leaving 11,898 records for title and abstract screening. The authors identified 241 records for full-text review, and 33 studies fulfilled all eligibility criteria and were included in the review. 

### 3.2. Characteristics of Included Studies

Of the thirty-three studies included in this review, fifteen studies originated from Asia, twelve from the EU, five from Northern America, and one from Russia. The majority of the studies were of a case-control design (29/34), while the remaining five were cohort studies. We did not identify any randomized clinical trials. The cases were compared to matched controls in nine studies, unmatched healthy controls in thirteen studies, unmatched patients with benign diseases in five studies, and the remaining six studies were made up of the non-cancer patients from the cohort studies and one study with a control group consisting of healthy subjects, benign diseases and prostate cancer [29]. The number of cases reported ranged from 13 [30] to 188 [29], while the number of controls ranged from 11 [31] to 155 [29]. The detailed study characteristics can be viewed in Table 1.

### 3.3. Sample Type and Analysis Method

The analysis details are summarized in Table 2. The majority of the studies (28/33) used plasma for detecting ctDNA; three studies tested both plasma and serum [32,35,39], one study used serum only [44], and one study used citrate plasma [59]. Usadel, 2002 [32], analyzed the APC marker on 15 paired samples and reported a higher sensitivity for plasma versus serum, although not significantly different (93% versus 40%, respectively, *p* = 0.08). Gao, 2015 [39], analyzed all markers on both plasma and serum, and they reported higher sensitivity in the serum cohort (RASSF1A 43.1% versus 52.5% and APC 24.1% versus 42.5% for plasma versus serum, respectively). The main method applied by the studies was quantitative methylation-specific PCR (QMSP, 27/33); three studies used digital PCR [46,50,56] and two studies used sequencing-based assays [55,59]. One study employed a method where the target DNA was increased by PCR and then quantified by Surface Enhanced Raman Spectroscopy [49]. 

### 3.4. Diagnostic Performance of Methylated Circulating Tumor DNA

The 33 included studies investigated a total of 40 different genes (see the comprehensive list in Appendix A). The three most frequently used genes were SHOX2, RASSF1A, and APC, which were all reported in seven independent cohorts (Figure 2). The sensitivity ranged from 8% to 93%, while the specificity ranged from 69% to 100%. Since there was substantial heterogeneity between the studies regarding biological material and choice of analysis method, we chose not to include pooled measures for sensitivity and specificity in the forest plots. The complete contingency tables for all studies can be accessed in Appendix A.

Pooling all genes from all independent cohorts resulted in 94 unique data points, as visualized in the hierarchical summary receiver operating characteristics (HSROC) curve (Figure 3). The summary sensitivity estimate was 46.9% (95% CI 41.0–52.9%), and the summary specificity estimate was 92.9% (95% CI 90.3–94.8%), suggesting good diagnostic properties of methylated ctDNA in lung cancer detection, especially in specificity. The summary diagnostic odds ratio was 11.5 (95% CI 8.6–15.4). The area under the HSROC curve (AUROC) was 0.81 (95% CI 0.77–0.84). 

### 3.5. Quality Assessment and Risk of Bias

The included studies were generally of acceptable quality; however, all studies lacked information in at least one area of the quality assessment (Figure 4). Only 4/33 studies explicitly stated that a consecutive or random sample of patients was enrolled, and only 3/33 studies avoided a case-control design. Consequently, patient selection posed the biggest potential risk for introducing bias. There was generally low concern that the included patients did not match the review question, with 3/33 studies considered as high-risk. One study had missing data on both cases and controls [29], the second study had a control group consisting entirely of males [23], and the third study included only adenocarcinoma patients [50].

Only 2/33 studies stated that the index test was performed blinded to the results of the reference standard, while 10/33 studies described the use of a pre-specified threshold or a training-validation design. The majority of the studies (19/33) did not sufficiently describe how the index test was performed or interpreted and was, therefore, evaluated as unclear in regard to the risk of bias. All studies were considered to have index tests within the scope of the review question.

The choice, conduct and interpretation of the reference standard were largely considered in low risk of introducing bias, and we found the target condition defined by the reference standard to match the review question. One study [29] was deemed unclear in terms of reference standards.

The interval between the index test and reference standard was not appropriate in three studies where the sample storage time was too long, while the timing aspect was not sufficiently described in 18/33 studies. The complete quality assessments can be found in the Appendix A.

The majority of the studies (25/33) were funded by public or non-profit organizations; four studies did not report on funding, and four studies were entirely or partly funded by a company or corporate funding source. Likewise, 24/33 studies reported that the authors had no potential conflicts of interest; three studies did not report whether conflicts of interest were present, three studies reported conflicts not pertaining to the funding source, and three studies reported conflicts of interest involving study funding.

There was no significant risk of publication bias in the present systematic review (Deek’s Funnel Plot Asymmetry Test *p* = 0.30, see Appendix A).

## 4. Discussion

The current review identified 33 separate studies. The same cohorts were used in more studies, leaving 31 unique cohorts addressing the question of whether methylated ctDNA in the blood is useful for identifying lung cancer. In total, 40 separate genes were investigated, and the sensitivity and specificity among the three most frequently analyzed genes (SHOX2, RASSF1A, and APC) varied from 8% to 93% and 69% to 100%, respectively. The summary sensitivity was 46.9%, and the summary specificity was 92.9%.

The accuracy requirements for a test depend heavily on the condition and intended clinical setting. A test with high specificity and low sensitivity may be used to rule in a diagnosis, while the opposite scenario may be used to rule out a diagnosis [60]. The summary sensitivity estimate from the present meta-analysis is very modest, while the summary specificity estimate is moderate to good. However, some of the genes investigated in the individual studies have performance measures that indicate a potential for future clinical application, possibly combined in biomarker panels. Methylated ctDNA in blood seems, at present, best suited as a rule-in tool.

Most often, the studies used a case-control design (28/33 studies). This type of design bears clear advantages; the possibility to conduct a retrospective study with blood samples already at hand and the option of matching the control group to the case group on various parameters. However, there is a higher risk of introducing bias in a retrospective study design, although this is expected to be less of an issue since patient-reported data were not included. This is reflected in the study quality assessments, where the five cohort studies were all judged as having a low risk of bias in the patient selection area, which was true for only four of the case-control studies. Prospective cohort studies are more costly and time-consuming but offer a better level of evidence [61]. We did not identify any randomized clinical trials on this subject.

The large majority of investigations were performed using plasma. The optimal biologic sample type for ctDNA detection has been a recurrent topic of discussion, but there is generally a consensus that plasma is preferable to serum [62]. This is partly due to a higher yield of ctDNA in plasma and partly due to a higher level of contamination with genomic DNA in serum [62,63]. Two studies included in the present review did a direct comparison between plasma and serum and found diverging results [32,39]. Usadel, 2002 [32], reported a non-significant difference in favor of plasma. Gao, 2015 [39], found serum to be superior in terms of sensitivity; however, they did not perform any statistical testing. This observed difference might simply be by chance since not all subjects were matched, and results from the 26 matched patients generated slightly different sensitivity and specificity. 

To meet the inclusion criteria, the reference test for diagnosing lung cancer should involve tumor cytology or histopathology. However, some of the studies included in the analysis did not consistently specify whether the diagnosis was obtained through cytological or histological biopsy or through surgical resection. Despite the lack of detailed information on the diagnostic method in all studies, the histological type of lung cancer was described, indicating that the diagnosis was confirmed rather than solely based on clinical suspicion. Reporting both histology and stage of the lung cancer cases is crucial, and future research should consider reporting diagnostic sensitivity and specificity for these subgroups provided sufficient participant numbers. 

One of the criteria for inclusion in the current study was the quantitative measurement of methylated DNA. In total, 27/33 studies performed QMSP. Probably this is because of the relatively low costs of equipment and assays and the ease of performance [64]. Both QMSP and digital PCR with MethyLight has demonstrated a strong correlation between the expected and observed methylation values, while NGS tended to overestimate the methylation level [18]. However, digital PCR is a much more sensitive analysis, as demonstrated by Yu and colleagues, who reported a 20-fold lower limit of detection with droplet digital PCR compared to conventional quantitative PCR [65]. A large, interlaboratory study concluded that droplet digital PCR could perform highly reproducible absolute quantification of a specific DNA target with a reported inter-laboratory difference of <12% [66]. Digital PCR is a more recently developed technology, which is reflected in the present review, where the oldest study performing digital PCR was from 2019 [46]. 

Pyrosequencing was employed by Zhang, 2022 [59] and Kim, 2022 [55]. The sequencing-based methods for quantifying methylated tumor DNA can be more time-consuming and costly compared to PCR-based methods. However, these methods can be used for genome-wide coverage with comparable sensitivity to digital PCR [67], while the PCR-based methods are limited to a single gene or smaller multiplex gene panels. The eligibility criteria specified that only studies reporting outcome data for single genes would be included in the review. This was the main reason why only two of the included studies performed sequencing since sequencing outcomes are often reported as large algorithms or multiplex panels. A recent review of the data on methylated ctDNA in ovarian cancer concluded that panels of methylation markers performed better than single genes [68]. The results from the present review may guide the choice of biomarkers to include in future multiplex panels for diagnosing lung cancer. 

The studies originated from different parts of the world. The majority of the studies (15/33) originated from China or other Asian countries, while twelve originated from EU countries and five were from North America. These geographical differences may impact the applicability of the results since studies have shown that a significant proportion of lung cancer cases among East Asian populations occur in individuals who are non-smokers, in contrast to Caucasians [69], and that DNA methylation is highly affected by smoking [70]. The studies included in this meta-analysis have not consistently reported race and a direct comparison of the diagnostic accuracy of the respective genes in different races is not possible. 

We applied wide criteria in the search strategy, searched the major databases and identified a very large number of potentially eligible studies. However, we can never be entirely certain that we have identified all relevant studies. Some studies may only have used the gene name and not any of the broader ctDNA terms in the title and abstract and thus would not be included in our search results. We did not search the gray literature or unpublished results. 

This review and meta-analysis combine data from multiple studies, allowing for a larger sample size and increased statistical power. This can enhance the precision and reliability of the findings. By pooling data from various studies, the meta-analysis provides a comprehensive overview of the available evidence on the diagnostic accuracy of methylated DNA in lung cancer. It helps identify consistent patterns or trends across different studies. The meta-analysis incorporates data from multiple populations and settings, possibly providing a more generalizable estimate of the diagnostic accuracy. This enhances the applicability of the findings to different patient populations. However, caution is advised when interpreting the summary estimates, as the list of investigated genes is extensive and diverse.

The quality of the individual studies included in the meta-analysis varied. Limitations or biases in the design, conduct, or reporting of the primary studies can potentially affect the validity and reliability of the meta-analysis results. Given the high number of case-control studies with healthy individuals in the control group, the results are subject to spectrum bias. A more accurate effect estimate may be obtained in cohort studies or case-control studies, including patients with benign diseases. In addition, there is a risk of publication bias, as studies with positive or significant results are more likely to be published, while studies with negative or non-significant findings may remain unpublished. This can introduce a bias in the overall estimate of diagnostic accuracy. However, as described previously, no significant risk of publication bias was found using Deek’s funnel plot asymmetry test. Variations in study characteristics such as patient and control group populations, assay methods, cutoff values, and study designs can introduce heterogeneity across studies. This heterogeneity can affect the pooling of results and warrants caution in the generalizability of the findings.

## 5. Conclusions

In conclusion, methylated ctDNA from blood samples can be detected by various methods, but the summary sensitivity estimate implies that further improvements are needed before this type of biomarker is ready for testing in a randomized clinical trial setting, much less for clinical implementation. The detailed results presented here may aid the selection of genes for biomarker panels. Future studies should consider the following points: (I) Adopt a cohort design to reduce the risk of bias; (II) Follow best practice guidelines for the preanalytical work-flow as suggested by Meddeb and colleagues [62] to reduce variability; (III) Rigorously follow the Standards for Reporting Diagnostic accuracy studies (STARD) guidelines [71] since all studies had at least one unclear item in the QUADAS-2 evaluation; (IV) Emphasize the detection of early-stage lung cancer, as this is where the survival benefit is most significant, and provide specific results for lung cancer stage and histology whenever possible. 

## Figures and Tables

**Figure 1 cancers-15-03959-f001:**
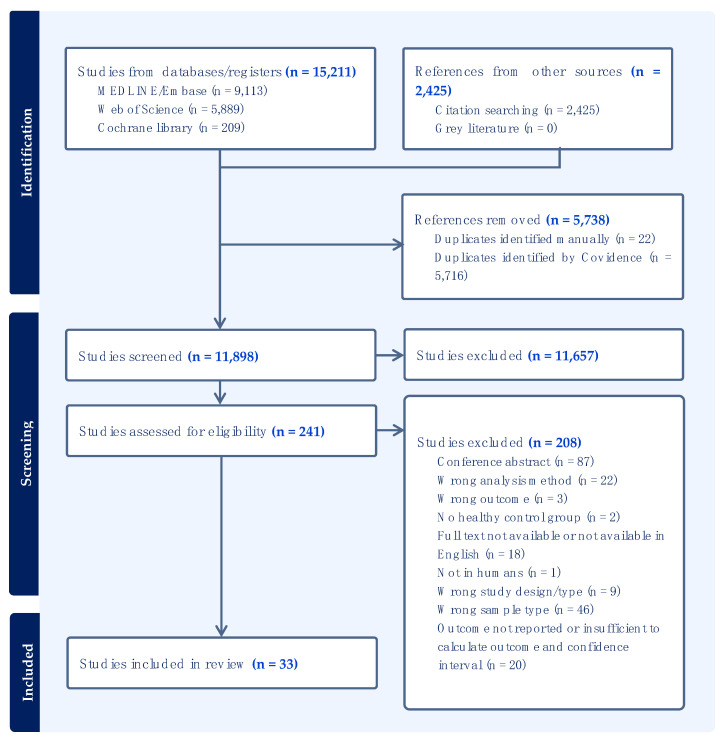
PRISMA flow chart.

**Figure 2 cancers-15-03959-f002:**
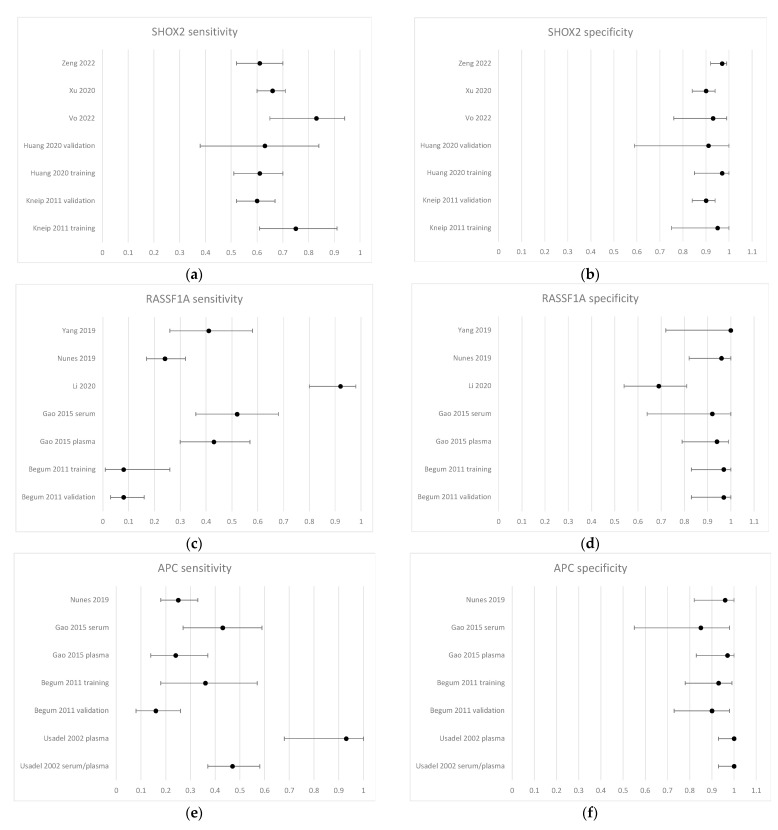
Forest plots of biomarker sensitivity. Forest plots of sensitivity (left-hand panels) and specificity (right-hand panels) for the three most frequently investigated genes. SHOX2 (**a**,**b**) was evaluated by Kneip, 2011 [29], Huang, 2020 [48], Vo, 2022 [57], Xu, 2020 [51], and Zeng, 2022 [58]. RASSF1A (**c**,**d**) was evaluated by Begum, 2011 [35], Gao, 2015 [39], Li, 2020 [49], Nunes, 2019 [45], and Yang, 2019 [31]. APC (**e**,**f**) was evaluated by Usadel, 2002 [32], Begum, 2011 [35], Gao, 2015 [39], and Nunes, 2019 [45]. Dots represent the estimated effect size, and error bars illustrate the 95% confidence intervals.

**Figure 3 cancers-15-03959-f003:**
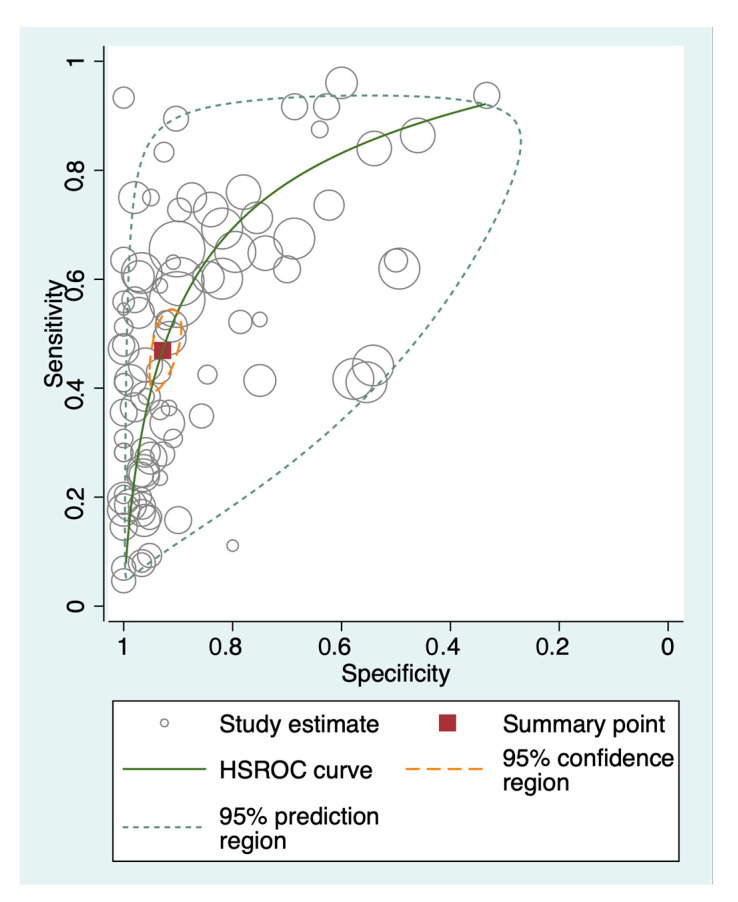
Hierarchical summary receiver operating characteristics curve. Hierarchical summary receiver operating characteristics (HSROC) curve of the 94 unique data points obtained from the 33 studies included in the meta-analysis. This is a graphical depiction of the random-effects model that includes estimates of the between-study variance. The open circles represent the study estimates, i.e., each target gene investigated in an independent cohort. The red square represents the summary point of all 94 data points, with the 95% confidence region outlined in yellow and the 95% prediction region outlined in gray. The solid green line is the summary HSROC curve.

**Figure 4 cancers-15-03959-f004:**
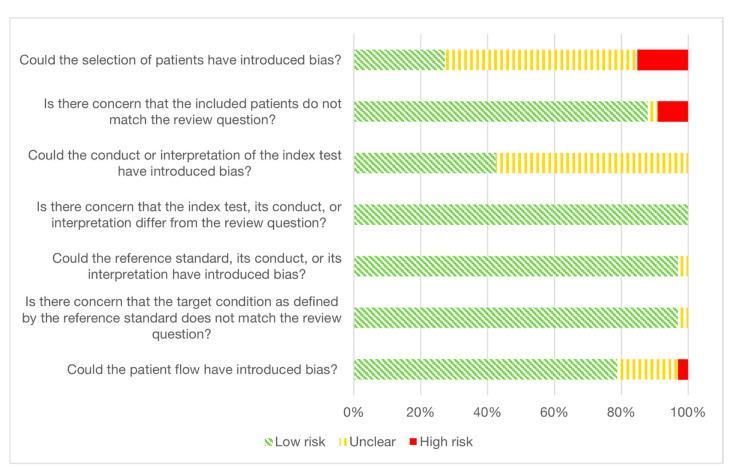
QUADAS-2. Stacked bar chart of the Quality Assessment of Diagnostic Accuracy Studies 2 (QUADAS-2) consensus judgments. Green diagonal stripes, yellow vertical stripes, and solid red areas represent the percentage of studies judged as low risk, unclear risk, and high risk of bias, respectively.

**Table 1 cancers-15-03959-t001:** Study characteristics.

Study ID	Region	Study Design	Cases	Histology	Stage	Controls	Cohort	Number of Cases	Number of Controls	Reference Standard
Usadel, 2002 [32]	Northern America	Case-control study	Retrospectively selected cases	LUSC 35/99 * (35%), LUAD 47/99 (48%), other 17/99 (17%)	I 53/99 * (54%), II 23/99 (23%), III 17/99 (17%), IV 6/99 (6%)	Unmatched healthy controls	Training	71 serum, 33 plasma (15 matched)	50	Histopathology or cytology.
Ostrow, 2009 [30]	Northern America	Case-control study	Retrospectively selected cases	LUSC 6/13 (46%), LUAD 1/13 (8%), other 6/13 (46%)	Not reported	Matched on certain characteristics	Training	13	24	Tumor tissue biopsy/histopathology
				LUSC 7/70 (10%), LUAD 47/70 (67%), other 16/70 (23%)	I 49/70 (70%), II 2/70 (3%), III 10/70 (14%), IV 4/70 (6%), no stage 5/70 (7%)		Validation	70	23 with nodules + 80 smokers with no nodules	
Zhang, 2010 A [33]	China	Case-control study	Retrospectively selected cases	LUSC 36/78 (46%), LUAD 30/78 (38%), other 12/78 (15%)	I–II 58/78 (74%), III–IV 20/78 (26%)	Unmatched healthy controls	Training	78	50	Histopathology or cytology
Zhang, 2010 B [34]	China	Case-control study	Retrospectively selected cases	LUSC 36/78 (46%), LUAD 30/78 (38%), other 12/78 (15%)	I–II 58/78 (74%), III–IV 20/78 (26%)	Unmatched healthy controls	Training	78	50	Tumor tissue biopsy/histopathology
Begum, 2011 [35]	Northern America	Case-control study	Retrospectively selected cases	LUSC 26/76 (34%), LUAD 36/76 (47%), other 14/76 (18%)	I 41/76 (54%), II 17/76 (22%), III 11/76 (14%), IV 5/76 (7%), unknown 2/76 (3%)	Matched on certain characteristics	Training	76	30	Histopathology or cytology
Kneip, 2011 [29]	EU	Case-control study	Retrospectively selected cases	LUSC 38/188 (20%), LUAD 31/188 (16%), SCLC 15/188 (8%), other/unknown 104/188 (55%)	I 37/188 (20%), II 29/188 (15%), III 53/188 (28%), IV 42/188 (22%), unknown 27/188 (14%)	Combination of healthy, benign and prostate cancer	Training	188	155	Histopathology or cytology
Ponomaryova, 2011 [23]	Other: Russia	Case-control study	Retrospectively selected cases	LUSC 34/52 (65%), LUAD 18/52 (35%)	I–II 25/52 (48%), III–IV 27/52 (52%)	Unmatched healthy controls	Training	52	26	Histopathology or cytology
Vinayanuwattikun, 2011 [36]	Other: Asian country	Case-control study	Retrospectively selected cases	NSCLC, not further described	The whole cohort was described as ‘advanced’.	Matched on certain characteristics	Training	38	52	Tumor tissue biopsy/histopathology
Balgkouranidou, 2014 A [37]	EU	Case-control study	Retrospectively selected cases	LUSC 23/44 ^#^ (52%), LUAD 20/44 (45%), missing 1/44 (2%)	I 14/44 ^#^ (32%), II–III 29/44 (66%), missing 1/44 (2%)	Unmatched healthy controls	Training	48	24 (same used for training and validation)	Histopathology or cytology
				LUSC 24/74 (32%), non-squamous 50/74 (68%)	IV 74/74 (100%)		Validation	74	24 (same used for training and validation)	
Powrozek, 2014 [38]	EU	Case-control study	Retrospectively selected cases	LUSC 20/70 (29%), LUAD 20/70 (29%), SCLC 23/70 (33%), other 7/70 (10%)	I 0/47 ^€^ (0%), II 7/47 (15%), III 23/47 (49%), IV 17/47 (36%)	Matched on certain characteristics	Training	70	100	Not described
Gao, 2015 [39]	China	Cohort study	Diagnostic work-up for LC	LUSC 23/58 (40%), LUAD 18/58 (31%), SCLC 2/58 (3%), other 15/58 (26%)	All were early-stage lung cancer (T1a–T2a)	Non-cancer participants who underwent diagnostic work-up	Training	58 plasma 40 serum	31 with benign disease, 23 healthy	Histopathology or cytology
Balgkouranidou, 2016 B [22]	EU	Case-control study	Retrospectively selected cases	LUSC 21/44 ^#^ (48%), LUAD 22/44 (50%), missing 1/44 (2%)	I 14/44 ^#^ (32%), II–III 29/44 (66%), missing 1/44 (2%)	Unmatched healthy controls	Training	48	49 (same used for training and validation)	Tumor tissue biopsy/histopathology
				LUSC 24/74 (32%), non-squamous 50/74 (68%)	IV 74/74 (100%)		Validation	74	49 (same used for training and validation)	
Powrozek, 2016 [40]	EU	Case-control study	Retrospectively selected cases	LUSC 20/65 (31%), LUAD 22/65 (34%), SCLC 19/65 (29%), other 4/65 (6%)	I 0/46 (0%), II 7/46 (15%), III 22/46 (48%), IV 17/46 (37%), limited 9/19 (47%), extensive 10/19 (53%)	Unmatched healthy controls	Training	65	95	Tumor tissue biopsy/histopathology
Powrozek, 2016 [41]	EU	Case-control study	Retrospectively selected cases	LUSC 30/70 (43%), LUAD 25/70 (36%), SCLC 15 (21%)	I 8/55 ^#^ (15%), II 12/55 (22%), III 19/55 (35%), IV 16/55 (29%)	Unmatched healthy controls	Training	70	80	Surgery specimen/histopathology
Aslam, 2017 [42]	Other: Asian country	Case-control study	Retrospectively selected cases	LUSC 19/34 (56%), LUAD 7/34 (21%), other 8/34 (24%)	Not reported	Matched on certain characteristics	Training	34	34	Tumor tissue biopsy/histopathology
Hulbert, 2017 [43]	Northern America	Cohort study	Diagnostic work-up for LC	LUSC 26/150 (17%), LUAD 121/150 (81%), other 3/150 (2%)	I 136/150 (91%), II 14/150 (9%), III 0/150 (0%), IV 0/150 (0%)	Non-cancer participants who underwent diagnostic work-up	Training	125	50	Surgery specimen/histopathology
Ooki, 2017 [44]	Northern America	Case-control study	Retrospectively selected cases	LUAD 43/43 (100%)	I 43/43 (100%)	Matched on certain characteristics	Training	43 LUAD	42 (same used for training and validation)	Histopathology or cytology
				LUSC 40/40 (100%)	I 40/40 (100%)		Validation	40 LUSC	42 (same used for training and validation)	
Nunes, 2019 [45]	EU	Case-control study	Retrospectively selected cases	LUSC 42/129 (33%), LUAD 65/129 (50%), SCLC 19/129 (15%), other 3/129 (2%)	I 15/129, II 11/129, III 27/129, IV 76/129	Non-cancer participants who underwent diagnostic work-up	Training	129	28	Histopathology or cytology
Villalba, 2019 [46]	EU	Case-control study	Retrospectively selected cases	LUSC 38/89 (43%), LUAD 51/89 (57%)	I 8/89 (9%), II 8/89 (9%), III 19/89 (21%), IV 52/89 (58%), missing 2/89 (2%)	Matched on certain characteristics	Training	89	25	Surgery specimen/histopathology
Yang, 2019 [31]	China	Cohort study	Diagnostic work-up for LC	LUSC 12/39 (31%), LUAD 25/39 (64%), other 2/39 (5%)	I 39/39 (100%)	Non-cancer participants who underwent diagnostic work-up	Training	39	11	Surgery specimen/histopathology
Chen, 2020 [47]	China	Cohort study	Diagnostic work-up for LC	LUSC 22/163 (13%), LUAD 139/163 (85%), other 2/163 (1%)	I 163/163 (100%)	Non-cancer participants who underwent diagnostic work-up	Training	163	83	Surgery specimen/histopathology
Huang, 2020 [48]	China	Cohort study	Diagnostic work-up for LC	LUSC 15/104 (14%), LUAD 53/104 (51%), SCLC 3/104 (3%), other 1/104 (1%), missing 32/104 (31%)	I 48/104 (46%), II 15/104 (14%), III 20/104 (19%), IV 21/104 (20%)	Unmatched patients with benign diseases	Training	104	36 with benign disease, 50 healthy	Surgery specimen/histopathology
				LUSC 4/19 (21%), LUAD 14/19 (74%), other 1/19 (5%)	I 12/19 (63%), II 4/19 (21%), III 3/19 (16%)		Validation	19	11	
Li, 2020 [49]	China	Case-control study	Retrospectively selected cases	LUSC 24/48 (50%), LUAD 18/48 (38%), other 6/48 (13%)	I–II 15/48 (31%), III–IV 33/48 (69%)	Unmatched healthy controls	Training	48	51	Histopathology or cytology
Wen, 2020 [50]	EU	Case-control study	Retrospectively selected cases	LUAD 48/48 (100%)	III 3/48 (6%), IV 45/48 (94%)	Unmatched healthy controls	Training	48	100	Histopathology or cytology
Xu, 2020 [51]	China	Case-control study	Retrospectively selected cases	LUSC 28/302 (9%), LUAD 236/302 (78%), SCLC 32/302 (11%), other 6/302 (2%)	I 68/302 (23%), II 62/302 (21%), III 72/302 (24%), IV 100/302 (33%)	Matched on certain characteristics	Training	302	153	Not described
Mastoraki, 2021 [52]	EU	Case-control study	Retrospectively selected cases	LUSC 19/48 (40%), LUAD 28/48 (58%), other 1/48 (2%)	I–II 28/48 (58%), III–IV 13/48 (27%), missing 7/48 (15%)	Matched on certain characteristics	Training	48 early stage	60 (same used for training and validation)	Histopathology or cytology
				Not available	IV 91/91 (100%)		Validation	91 stage IV	60 (same used for training and validation)	
Park, 2021 [53]	Other: Asian country	Case-control study	Retrospectively selected cases	Not available	Not available	Unmatched healthy controls	Training	64	64	Tumor tissue biopsy/histopathology
Szczyrek, 2021 [54]	EU	Case-control study	Diagnostic work-up for LC	LUSC 34/101 (34%), LUAD 52/101 (51%), SCLC 8/101 (8%), other 7/101 (7%)	IA–IIIA 27/101 (27%), IIIB–IV 66/101 (65%), missing 8/101 (8%)	Unmatched healthy controls	Training	101	45	Tumor tissue biopsy/histopathology
Kim, 2022 [55]	Other: Asian country	Case-control study	Diagnostic work-up for LC	LUSC 30/72 (42%), LUAD 31/72 (43%), other 11/72 (15%)	I 41/72 (57%), II 26/72 (36%), III 3/72 (4%), IV 2/72 (3%)	Unmatched patients with benign diseases	Training	72	61	Surgery specimen/histopathology
Palanca-Ballester, 2022 [56]	EU	Case-control study	Retrospectively selected cases	LUSC 13/44 (30%), LUAD 31/44 (70%)	I 4/44 (9%), II 7/44 (16%), III 3/44 (7%), IV 30/44 (68%)	Unmatched patients with benign diseases	Training	44	39	Other: Histopathology or cytology
Vo, 2022 [57]	Other: Asian country	Case-control study	Retrospectively selected cases	Not available	I 2/30 (7%), II 8/30 (27%), III 15/30 (50%), IV 5/30 (17%)	Unmatched healthy controls	Training	30	27	Other: Histopathology or cytology.
Zeng, 2022 [58]	China	Case-control study	Retrospectively selected cases	LUSC 58/121 (48%), LUAD 63/121 (52%)	I–II 78/121 (64%), III–IV 43/121 (36%)	Unmatched patients with benign diseases	Training	121	121	Surgery specimen/histopathology
Zhang, 2022 [59]	China	Case-control study	Retrospectively selected cases	LUSC 8/23 (35%), LUAD 10/23 (43%), SCLC 5/23 (22%)	I–II 2/23 (9%), III–IV 21/23 (91%)	Unmatched patients with benign diseases	Training	23	56	Histopathology or cytology

Characteristics of all studies included in the review. Studies are arranged according to year of publication, starting with the oldest study and then alphabetically if more studies were published in the same year. Study cohorts used in more than one publication are labeled with A, B, etc. Independent training and validation cohorts are reported in separate rows. Study ID consists of the first author’s last name and the year of publication, followed by the reference. Region refers to the geographical region in which the study was performed. LC, lung cancer; LUAD, lung adenocarcinoma; LUSC, lung squamous cell carcinoma; NSCLC, non-small cell lung cancer; SCLC, small cell lung cancer. * A subset of patients had blood samples collected, but data on histology and stage were only reported for the whole patient cohort. # The number of plasma samples was reported as *n* = 48; however, in the table reporting clinicopathological data for BRMS1 methylated, only *n* = 44 patients were reported. ^€^ The detailed disease stage was only reported for the NSCLC patients.

**Table 2 cancers-15-03959-t002:** Sample type and analysis method.

Study ID	Sample Type	Analysis Method	Assay Type	How Was the Cut-Off Determined?
Usadel, 2002 [32]	Plasma; Serum	QMSP	Single gene	Not reported
Ostrow, 2009 [30]	Plasma	QMSP	Single gene	Defined by a training cohort and validated in an independent cohort
Zhang, 2010 A [33]	Plasma	QMSP	Single gene	Not reported
Zhang, 2010 B [34]	Plasma	QMSP	Single gene	Not reported
Begum, 2011 [35]	Plasma; Serum	QMSP	Single gene	Defined by a training cohort and validated in an independent cohort
Kneip, 2011 [29]	Plasma	QMSP	Single gene	Defined by a training cohort and validated in an independent cohort
Ponomaryova, 2011 [23]	Plasma	QMSP	Single gene	Defined by a training cohort
Vinayanuwattikun, 2011 [36]	Plasma	QMSP	Single gene	Defined by a training cohort
Balgkouranidou, 2014 A [37]	Plasma	QMSP	Single gene	Not reported
Powrozek, 2014 [38]	Plasma	QMSP	Single gene	Defined in a previous study
Gao, 2015 [39]	Plasma; Serum	QMSP	Multiplex	Defined by a training cohort
Balgkouranidou, 2016 B [22]	Plasma	QMSP	Single gene	Not reported
Powrozek, 2016 [40]	Plasma	QMSP	Single gene	Defined by a training cohort
Powrozek, 2016 [41]	Plasma	QMSP	Single gene	Not reported
Aslam, 2017 [42]	Plasma	QMSP	Single gene	Not reported
Hulbert, 2017 [43]	Plasma	QMSP	Single gene	Defined by a training cohort
Ooki, 2017 [44]	Serum	QMSP	Single gene	Defined by a training cohort and validated in an independent cohort
Nunes, 2019 [45]	Plasma	QMSP	Multiplex	Defined by a training cohort and validated in an independent cohort
Villalba, 2019 [46]	Plasma	Digital PCR	Single gene	Defined by a training cohort
Yang, 2019 [31]	Plasma	QMSP	Single gene	Defined by a training cohort
Chen, 2020 [47]	Plasma	QMSP	Single gene	Defined by a training cohort
Huang, 2020 [48]	Plasma	QMSP	Not described	Defined by a training cohort and validated in an independent cohort
Li, 2020 [49]	Plasma	PCR-SERS	Single gene	Defined by a training cohort
Wen, 2020 [50]	Plasma	Digital PCR	Single gene	Defined by a training cohort
Xu, 2020 [51]	Plasma	QMSP	Multiplex	Arbitrarily set at 90% specificity for both markers.
Mastoraki, 2021 [52]	Plasma	QMSP	Single gene	Defined by a training cohort and validated in an independent cohort
Park, 2021 [53]	Plasma	QMSP	Single gene	Defined by a training cohort
Szczyrek, 2021 [54]	Plasma	QMSP	Single gene	Defined by a training cohort
Kim, 2022 [55]	Plasma	Pyrosequencing	Single gene	Defined by a training cohort and validated in an independent cohort
Palanca-Ballester, 2022 [56]	Plasma	Digital PCR	Single gene	Defined by a training cohort
Vo, 2022 [57]	Plasma	QMSP	Single gene	Defined by a training cohort
Zeng, 2022 [58]	Plasma	QMSP	Single gene	Defined by a training cohort
Zhang, 2022 [59]	Plasma or serum	Pyrosequencing	Single gene	Not reported

Overview of the sample types and analysis methods employed by the included studies. Studies are arranged according to year of publication, starting with the oldest study and then alphabetically if more studies were published in the same year. Study cohorts used in more than one publication are labeled with A, B, etc. Study ID consists of the first author’s last name and the year of publication, followed by the reference. PCR, polymerase chain reaction; QMSP, quantitative methylation-specific polymerase chain reaction; SERS, surface-enhanced Raman spectroscopy.

## Data Availability

The data presented in this study are available in the article and the Appendix A.

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
