# Peer review of "Methylated Circulating Tumor DNA in Blood as a Tool for Diagnosing Lung Cancer: A Systematic Review and Meta-Analysis"

_cancers, 2023, doi:10.3390/cancers15153959_

Round 1

Reviewer 1 Report

  • A brief summary: Hornemann Borg et al. conducted an extensive literature review, summarizing current evidence on using methylated circulating tumor DNA as a diagnostic biomarker for diseases, including lung cancer. The review aims to assist future research in this field.
  • General concept comments

ü  Specific comments:

o    Can the sample size be increased in the current study to reduce risk of bias?

o    Other studies have already shown that ctDNA act as potential biomarkers against cancers such as CRC, so I would like to learn the uniqueness of the review report submitted by the authors.

o    Furthermore, it would be good to mention biological therapeutic significance of the study related to ctDNA only in cancer cells vs normal cells.

ü  The review is well written in fluent English, highly informative and suggest its acceptance in Cancers.

Author Response

Specific comments:

  1. Can the sample size be increased in the current study to reduce risk of bias?

Reply: We thank you for providing us with a helpful review report and insightful comments.

We acknowledge the potential benefits of increasing the sample size in our current study to reduce the risk of bias. We could, theoretically, increase the number of studies included in the review. However, we also recognize that doing so might require loosening our inclusion and exclusion criteria, which could compromise the robustness of the review and change the research question.

Changes: None.

  1. Other studies have already shown that ctDNA act as potential biomarkers against cancers such as CRC, so I would like to learn the uniqueness of the review report submitted by the authors.

Reply: Yes, ctDNA is also widely investigated in other cancer types such as colorectal cancer. We do not know of any recently published or planned systematic reviews investigating the use of methylated ctDNA for detection of lung cancer. One of the selection criteria in the present review was studies reporting diagnostic sensitivity and specificity for single-genes and not only multiplex panels or combined biomarkers. This criterion was put in to increase the usefulness of the review for researchers who want to select markers for future studies e.g., for multiplex panels. We do believe that the present review provides new insight by carefully collecting and analyzing the existing data in this area.

Changes: None.

  1. Furthermore, it would be good to mention biological therapeutic significance of the study related to ctDNA only in cancer cells vs normal cells.

Reply: We are not certain about this comment and would prefer if the reviewer could elaborate.

Changes: None.

Reviewer 2 Report

Drs. Borg and colleagues present an article on circulating tumor DNA in blood as a tool for diagnosing lung cancer. This review is well structured. The authors have analyzed several studies and finally came up with 33 reports, which had sufficient data to be analyzed. The authors concluded, that more work is needed before the analysis of methylated ctDNA can be used as a tool for cancer diagnosis. There were problems about sensitivity (46%), whereas when methylated DNA was detected the specificity was high.

There are a few aspects the authors should include in their review:

1. put more emphasis on selecting more than 40 genes into such an analysis

2. As metabolic changes play a major role in early cancer development, genes playing a role should be included in future studies

3. Staging of the cases in these 33 studies should be included into the review, as this might have an impact: shedding of tumor DNA is low in early lung cancer, whereas quite common in stages IIIA/B and IV. But in these cases obtaining tumor tissue for example by EBUS is not complicated. In early stages below IIB the analysis of methylated ctDNA might be more important. Here nodule(s) are detected by CT, and a positivity for ctDNA might help in the guidance of further evaluation

4. The authors might also come up with an analysis how far ctDNA might be used to sort pulmonary carcinomas into subtypes. RASSF1A might not be a gene separating adenocarcinomas from squamous ones.

The quality of English is OK and requires only minimal corrections

Author Response

  1. Put more emphasis on selecting more than 40 genes into such an analysis

Reply: We thank you for providing us with a helpful review report and insightful comments.

If we understand your comment correctly, we should emphasize the large amount of heterogeneity in the meta-analysis given that we pool results from 40 different genes to make the summary estimates. If this is what you mean, we agree and thank you for the observation. It is indeed a long and very varied list of genes, and the summary estimates should be interpreted very cautiously.

Changes: Page 18, lines 416-418.

  1. As metabolic changes play a major role in early cancer development, genes playing a role should be included in future studies

Reply: In the conclusion on page 19, lines 444-446, we advocate for additional studies exploring ctDNA's potential as an early-stage lung cancer marker, crucial for achieving survival benefits.

Changes: Page 19, lines 444-446.

  1. Staging of the cases in these 33 studies should be included into the review, as this might have an impact: shedding of tumor DNA is low in early lung cancer, whereas quite common in stages IIIA/B and IV. But in these cases obtaining tumor tissue for example by EBUS is not complicated. In early stages below IIB the analysis of methylated ctDNA might be more important. Here nodule(s) are detected by CT, and a positivity for ctDNA might help in the guidance of further evaluation

Reply: We agree that the disease stage is very important in order to evaluate the results, as clinicians are naturally most interested in biomarkers which can detect early-stage disease. We have therefore collected the available data on tumor stage as well as histology and included them in Table 1 (study characteristics).

However, due to limited data on histology and stage-specific results in most studies, the estimated values carry significant uncertainty. Future research should consider reporting diagnostic sensitivity and specificity for subgroups based on stage or histology, provided sufficient participant numbers. We have highlighted this proposal in the discussion section page 17, lines 367-369.

Changes: Table 1 now includes 'Histology' and 'Stage' columns. Page 17, lines 367-369.

  1. The authors might also come up with an analysis how far ctDNA might be used to sort pulmonary carcinomas into subtypes. RASSF1A might not be a gene separating adenocarcinomas from squamous ones.

Reply: Yes, that would certainly be both interesting and very useful. However, as described in the above reply most of the studies did not perform or report subgroup analyses based on histology and stage. Only a few studies did, so we were not able to make the analysis proposed by the reviewer.

Changes: See Table 1, columns ‘Histology’ and ‘Stage’ have been added.

Reviewer 3 Report

The systematic review and meta-analysis by Borg et al summarizes the main studies regarding the detection of lung cancer using DNA methylation in blood, namely plasma and serum samples. This type of summary of the studies currently publicated on the topic are important to understand which markers are the most reported and promising ones to further be evaluated in clinicals trial for lung cancer detection. The study is very well-written, well organized, clear and very interesting. I only have a few comments to further complete the work.

- One of the other important biological material obtained by a non-invasive manner described before for lung cancer detection is sputum. I think it would be interesting for the authors to add a section reporting the studies using sputum and which are the more relevant methylated genes for lung cancer detection.

- Another aspect to have into account is the lung cancer subtypes. The authors should add a section with the studies that describe differently methylated genes between the  lung cancer subtypes and which ones are more important for the detection of non-small cell lung cancer vs small-cell lung cancer, for example. 

By adding these new information, I think the study will further improve its novelty and interest for the scientific community.

Author Response

  1. One of the other important biological material obtained by a non-invasive manner described before for lung cancer detection is sputum. I think it would be interesting for the authors to add a section reporting the studies using sputum and which are the more relevant methylated genes for lung cancer detection.

Reply: We thank you for providing us with a helpful review report and insightful comments.

It is certainly very important to investigate and collate the current evidence on the use of methylated ctDNA in sputum for detection of lung cancer. However, the number of studies could be substantial, and we find that we could do better justice to the subject in a separate literature review. Our group has previously published a systematic review concerning the use of methylated ctDNA in bronchial lavage fluid, and we plan to do similar work regarding sputum and pleural effusion fluid in the near future.

Changes: None.

  1. Another aspect to have into account is the lung cancer subtypes. The authors should add a section with the studies that describe differently methylated genes between the lung cancer subtypes and which ones are more important for the detection of non-small cell lung cancer vs small-cell lung cancer, for example. 

By adding these new information, I think the study will further improve its novelty and interest for the scientific community.

Reply: We agree that the disease stage is very important in order to evaluate the results, as clinicians are naturally most interested in biomarkers which can detect early-stage disease. We have therefore collected the available data on tumor stage as well as histology and included them in Table 1 (study characteristics).

However, due to limited data on histology and stage-specific results in most studies, the estimated values carry significant uncertainty. Future research should consider reporting diagnostic sensitivity and specificity for subgroups based on stage or histology, provided sufficient participant numbers. We have highlighted this proposal in the discussion section page 17, lines 367-369.

Changes: Table 1 now includes 'Histology' and 'Stage' columns. Page 17, lines 367-369.